# Towards Personalised Contrast Injection: Artificial-Intelligence-Derived Body Composition and Liver Enhancement in Computed Tomography

**DOI:** 10.3390/jpm11030159

**Published:** 2021-02-24

**Authors:** Daan J. de Jong, Wouter B. Veldhuis, Frank J. Wessels, Bob de Vos, Pim Moeskops, Madeleine Kok

**Affiliations:** 1Department of Radiology, University Medical Center Utrecht, Heilberglaan 100, 3584 CX Utrecht, The Netherlands; d.j.dejong4@students.uu.nl (D.J.d.J.); W.Veldhuis@umcutrecht.nl (W.B.V.); f.j.wessels-3@umcutrecht.nl (F.J.W.); 2Quantib-U, Padualaan 8, 3584 CH Utrecht, The Netherlands; b.devos@quantib.com (B.d.V.); p.moeskops@quantib.com (P.M.)

**Keywords:** computed tomography, artificial intelligence, contrast media, body composition

## Abstract

In contrast-enhanced computed tomography, total body weight adapted contrast injection protocols have proven successful in achieving a homogeneous enhancement of vascular structures and liver parenchyma. However, because solid organs have greater perfusion than adipose tissue, the lean body weight (fat-free mass) rather than the total body weight is theorised to cause even more homogeneous enhancement. We included 102 consecutive patients who underwent a multiphase abdominal computed tomography between March 2016 and October 2019. Patients received contrast media (300 mgI/mL) according to bodyweight categories. Using regions of interest, we measured the Hounsfield unit (HU) increase in liver attenuation from unenhanced to contrast-enhanced computed tomography. Furthermore, subjective image quality was graded using a four-point Likert scale. An artificial intelligence algorithm automatically segmented and determined the body compositions and calculated the percentages of lean body weight. The hepatic enhancements were adjusted for iodine dose and iodine dose per total body weight, as well as percentage lean body weight. The associations between enhancement and total body weight, body mass index, and lean body weight were analysed using linear regression. Patients had a median age of 68 years (IQR: 58–74), a total body weight of 81 kg (IQR: 73–90), a body mass index of 26 kg/m^2^ (SD: ±4.2), and a lean body weight percentage of 50% (IQR: 36–55). Mean liver enhancements in the portal venous phase were 61 ± 12 HU (≤70 kg), 53 ± 10 HU (70–90 kg), and 53 ± 7 HU (≥90 kg). The majority (93%) of scans were rated as good or excellent. Regression analysis showed significant correlations between liver enhancement corrected for injected total iodine and total body weight (*r* = 0.53; *p* < 0.001) and between liver enhancement corrected for lean body weight and the percentage of lean body weight (*r* = 0.73; *p* < 0.001). Most benefits from personalising iodine injection using %LBW additive to total body weight would be achieved in patients under 90 kg. Liver enhancement is more strongly associated with the percentage of lean body weight than with the total body weight or body mass index. The observed variation in liver enhancement might be reduced by a personalised injection based on the artificial-intelligence-determined percentage of lean body weight.

## 1. Introduction

Even if ultrasound represents the first-line technique for the assessment of liver structure and potential lesions [1], contrast-enhanced computed tomography (CT) is commonly used to detect and characterise liver lesions [2,3]. The majority of these lesions are hypovascular and are, therefore, better identifiable with portal venous contrast enhancement [4,5]. A minimum enhancement of liver tissue of 50 HU is considered essential to ensure appropriate detectability [6,7,8]. The degree of contrast enhancement in CT is dependent on different factors: CT scan parameters (e.g., tube voltage, scan delay), injection parameters (e.g., amount of injected iodine), and patient-related factors (e.g., height, weight, cardiac output) [9]. The most widespread practise is to administer iodine contrast in fixed-contrast media injection protocols. Fixed protocols result in varying enhancement levels because of differences in body size and composition [9]. Lowering the dose of contrast media decreases the sensitivity and specificity in the detection and characterisation of liver lesions [10]. Higher doses of contrast media are costly and might increase the risk of renal toxicity [11,12]. A personalised protocol for iodine dosing should be preferred to the standard fixed-contrast protocol [13]. In this respect, body-weight-adapted contrast injection protocols have proven successful in achieving a more homogeneous enhancement of vascular structures and liver parenchyma in patients [8,14,15,16,17]. However, total body weight (TBW) is not the only relevant body-size-related factor; lean body weight (LBW) and body mass index (BMI) might also be important. Solid organs have greater perfusion than adipose tissue [18]; consequently, using LBW (or the fat-free mass) as the basis for determining the amount of iodine is hypothesised to result in more uniform liver enhancement than using TBW or BMI [18,19].

Some previous studies concluded that injection protocols based on LBW rather than on TBW alone performed better in terms of liver enhancement [13,18,19,20]. However, we find these results not to be generalisable to our clinic because many of the aforementioned studies were performed in populations with smaller ranges in weight.

Furthermore, these studies did not use body composition on a per patient basis, but performed analysis on averaged body composition values [13,19] or estimated the body composition using empirically derived formulas [18,20].

We want to take personalised medicine a step further, using artificial intelligence as a way to determine body composition. We will use a tool that automatically segments clearly visible structures such as fat, muscle, and bone on scanned images and determines the body composition of a patient. The automated nature of this technique makes it possible to dose contrast material in real-time and in a personalised fashion, and may have wide implications.

In this study, we retrospectively evaluated the influence of TBW, BMI, and artificial-intelligence-derived LBW on liver enhancement in multiphase abdominal CT, showing that subjective image quality was related to liver enhancement.

## 2. Materials and Methods

### 2.1. Patients

We retrospectively included patients from the period of March 2016 to October 2019. We included the first CT scan of all patients who underwent a multiphase abdominal CT, including an unenhanced CT for suspicion of a kidney tumour, on a spectral CT scanner in the University Medical Center Utrecht. Inclusion criteria were an age of 18 years or older and known patient weight and height. Based on these criteria, we identified 122 patients. Exclusion criteria were patients with liver cirrhosis (*n* = 2), a fatty liver (<40 HU) (*n* = 12), numerous liver metastases (*n* = 1), a partial hepatectomy (*n* = 2), and technical problems during CT examination (*n* = 1), leaving a study population of 102 patients. The Dutch Law on Medical Research (WMO) did not apply to this retrospective cohort study according to the local medical ethical committee (METC, ref. 20-025/C). No informed consent was obtained given the anonymous research data handling.

### 2.2. Imaging Protocols

All included multiphase CTs were performed on a spectral CT scanner (IQon Spectral CT, Philips Healthcare, Best, The Netherlands). The scan range for the unenhanced and arterial phase was the upper abdomen. The scan range for the portal venous phase was set from approximately 1 cm cranial of the diaphragm to the lower pelvis. The scan range for the (possible) equilibrium phase was set from the kidneys to just caudal of the bladder.

Scans were performed with the following parameters: tube voltage 120 kV, 64 × 0.625 mm collimation, gantry rotation time of 0.27 s, and tube current was switched on with a quality reference tube current of 116 mAs. Image reconstruction was performed in the axial plane for the unenhanced and arterial phase, with 3 and 5 mm slice thicknesses and 2 and 4 mm increments. Image reconstruction was performed in the axial, coronal, and sagittal plane for the portal venous phase, with 5 mm slice thicknesses and 4mm increments. All images were reconstructed using a B (abdominal) kernel at iDose level 3. 

All scans were performed with bolus tracking. A circular region of interest (ROI) was placed in the abdominal aorta with a threshold of 150 HU. The post-threshold delay before scanning was 20 s for the arterial phase and 90 s for the portal venous phase.

### 2.3. Contrast Material Injection and CT Protocols

All patients received an 18–20 G cannula in an antecubital vein before injection. Preheated iodinated contrast (Ultravist, Iopromide 300 mgI/mL; Bayer Healthcare, Berlin, Germany) was injected using a standard dual-head CT power injector (Stellant, Bayer Healthcare, Berlin, Germany). The contrast media was preheated to 37 °C to decrease viscosity [21].

In current clinical practice, body-weight-adapted protocols are used for the multiphase abdominal CT. Injection parameters were divided into three different weight groups: ≤70 kg, 70–90 kg, and ≥90 kg. The total injected volume, iodine, and flow rate were: 120 mL, 36.0 gI, 4 mL/s for group ≤70 kg; 150 mL, 45.0 gI, 4.5 mL/s for group 70–90 kg, and 185 mL, 55.5 gI, 5 mL/s for group ≥90 kg, respectively. A saline flush of 50 mL followed the contrast bolus at the same flow rate. In some cases, technicians adapted the amount of contrast media according to their experience, which was recorded in the scan protocol. In further analysis, we did not analyse weight groups, but instead used the weight of the patient; therefore, changes in scan protocol had no effect on analyses.

### 2.4. Quantitative Image Analysis

The body composition was calculated with the Quantib-U bod composition algorithm [22] on unenhanced images (Figure 1) [23]. Firstly, using a convolutional neural network, the method automatically detected the slice at the third lumbar vertebra from the CT data set (resampled to 5mm slices). Secondly, this slice was automatically segmented into visceral fat, subcutaneous fat, psoas muscle, abdominal muscle, and long spine muscle using a second convolutional neural network. Using the areas of these segmentations in proportion to those of the entire slice, percentages of body composition were calculated. To minimise the influence of the exact slice that was selected, the areas were computed by segmenting a total of five slices around the detected L3 level—two above and two below—and averaging the results. The %LBW (percentage of lean body weight) was defined as 100%—% total body fat (=subcutaneous fat % + visceral fat %). Total fat and LBW in kilograms were then calculated using TBW. Moreover, %LBW is an areal measure and LBW is in kilograms. 

CT liver enhancement values (HU) were measured (M.K., who has seven years of experience in CT imaging) on the unenhanced and portal venous phase images using circular regions of interest (ROI) of 1–2 cm in diameter. ROIs were placed in three different liver segments (S2, S8, and S7) according to the Couinaud segmental classification and mean values were calculated (Figure 2). The degree of contrast enhancement in the liver was defined as the change in enhancement values (ΔHU) and was calculated by subtraction of the unenhanced values from post-contrast enhancement values.

### 2.5. Qualitative Image Analysis

The quality of all scans was independently graded by two radiologists (F.W. and M.K., with eleven and four years of experience in abdominal radiology, respectively) who were blinded to the injection protocols. The timing of the scans and the subjective liver enhancement were scored. For scan timing, a five-point scale was used to evaluate enhancement of the common portal vein (1 = too early (non-diagnostic); 2 = early (moderate, but still diagnostic); 3 = portal venous phase (good); 4 = late (moderate, but still diagnostic); 5 = too late (non-diagnostic)). Liver enhancement was assessed using a four-point Likert scale (1 = excellent; 2 = good; 3 = moderate but still diagnostic; 4 = non-diagnostic). We arbitrarily defined enhancements of >70 HU and <40 HU as non-diagnostic.

### 2.6. Statistical Analysis

Statistical analyses were performed in SPSS version 26 (SPSS Inc., Chicago, IL, USA). Normality was checked using histograms and the Shapiro-Wilk test. Continuous variables were reported as the mean with standard deviation (± SD) for normally distributed data and as the median with an interquartile range (IQR) for non-normal distributed data. Categorical variables were reported as proportions. Continuous variables with normal distributions were compared using the repeated measures ANOVA for dependent measures or a one-way ANOVA for independent measures. A Kruskal–Wallis test was used for non-parametric continuous variables. All tests were performed with post hoc comparison. The inter-rater variability was determined using Cohen’s kappa. All p-values were 2-sided and a *p*-value of less than 0.05 was considered to be statistically significant.

Enhancement parameters of the liver obtained for further analyses were changed into enhancement values per gram of iodine (ΔHU/gI). These enhancement values were subsequently adjusted for TBW or LBW in kilograms (ΔHU/(gI/TBW) and ΔHU/(gI/LBW)), according to a method proposed by Heiken et al. [8] and Kondo et al. [19]. We used %LBW on a per-patient basis. Both single- and multivariable linear regressions between TBW, BMI, and %LBW and changes in enhancement values per gram of iodine (ΔHU/gI) or the adjusted enhancement values ΔHU/(gI/TBW) and ΔHU/(gI/LBW) were evaluated (Appendix A).

### 2.7. Simulation of Future Potential Clinical Applicability

Based on the formed regression formulas, we analysed the potential impacts for future patients by assessing the amount of contrast media needed to reach sufficient liver enhancement using our regression formulas for both %LBW and TBW. Our calculations for sufficient enhancement were based on an increase of 50 HU in the portal venous phase [6,7,8].

## 3. Results

### 3.1. Baseline Characteristics 

The 102 patients (70.6% male) had a median age of 68 years (IQR: 57–74). Their median TBW was 81.0 kg (IQR: 72.8–90.0)—19.6% were below 70 kg and 19.6% were above 90 kg. The median %LBW was 49.8% (IQR: 35.8–55.3) and the mean BMI was 26.3 kg/m2 (SD: ±4.18). Patients in the group ≤ 70 kg received a median of 36.0 g (IQR: 36.0–43.5) of iodine, the group 70–90 kg received 45.0 g (IQR: 39.0–45.0), and the group ≥ 90 kg 45.0 g (IQR: 45.0–45.7). Overall, the patients received 42.6 g (SD: ±4.42) of iodine per scan (Table 1).

### 3.2. Quantitative Image Quality

Mean enhancement values in different liver segments were as follows: S2 54.3 HU (SD: ±5.83), S8 54.8 HU (SD: ±6.61), and S7 54.3 HU (SD: ±9.30). There was no significant difference in enhancement between the liver segments for all groups. The overall mean enhancement was 54.6 HU (SD: ±10.2; range: 25.0–93.3) and 28.4% did not reach the proposed enhancement of 50 HU or more. The mean enhancement value was for ≤70 kg 60.7 HU (SD: ±12.4), for 70-90 kg was 53.3 HU (SD: ±9.25), and for ≥90 kg was 52.4 HU (SD: ±7.45). The between-group difference reached significance (*p* = 0.007) and in post hoc analysis the ≤70 kg group was enhanced significantly more than the 70–90 kg group (*p* = 0.019) and ≥90 kg group (*p* = 0.034) (Table 2). The percentages of patients enhanced by <50 HU were 20%, 30%, 35% in the ≤70 kg, 80–90 kg, and ≥90 kg groups, respectively. The percentages of patients enhanced by >70 HU were 30%, 4.8%, 0.0% in the ≤70 kg, 80–90 kg, and ≥90 kg groups, respectively (Appendix A).

### 3.3. Qualitative Image Quality

The inter-rater variability was good for scan timing (*k* = 0.882 (95% CI: 0.825–0.920)) and liver enhancement (*k* = 0.921 (95% CI: 0.833–0.946)). For timing, no scans were found to be non-diagnostic (Appendix A). For liver enhancement, nearly all scans were of good (25.5%) or moderate (5.90%) quality, while one scan was non-diagnostic scored by only one of the observers (objective liver enhancement 25 HU) (Appendix A). Most scans of moderate quality scored lower than 40 HU.

### 3.4. Regression Analysis

For the association between liver enhancement values per gram of iodine (ΔHU/gI) and body parameters, a correlation was observed with TBW (*r* = 0.531; R2 = 0.282; *p* < 0.001), while no significant values were observed for BMI (*p* = 0.253) or %LBW (*p* = 0.493) (Appendix A). The formula for this relationship is: gI = ΔHU/(2.075-0.01 TBW), which can also be written as gI = ΔHU/(2.075-0.01 TBW) (Figure 3A). For the liver enhancement values additionally adjusted per gram of iodine per TBW (ΔHU/(gI/TBW)), no significant correlations were found (BMI; *p* = 0.139. TBW; *p* = 0.302. %LBW; *p* = 0.628) (Appendix A). For the liver enhancement values additionally adjusted per gram of iodine per LBW (ΔHU/(gI/LBW)) the strongest association was observed with %LBW (*r* = 0.733; R2 = 0.538; *p* < 0.001), no significant correlations were observed for BMI (*p* = 0.099) or TBW (*p* = 0.371) (Figure 3B) (Appendix A). The formula for this relationship is: ΔHU/(gI/LBW) = 10.3 + 0.823 %LBW or gI = ΔHU/(10.3/LBW + 82.3/TBW).

### 3.5. Simulation of Future Potential Clinical Applicability

For the 102 included patients, we used an average of 42.6 g of iodine (SD: ±4.42; range: 36–55.5) per scan in the standard protocol, totaling approximately 4345 g of iodine and 14.5 litres of contrast for 102 patients. This is on average 0.532 g (SD: ±0.0811; range: 0.33–0.75) of iodine per kilogram TBW. For our regression formula based on %LBW, an estimated average of 39.4 g (SD: ±6.05; range: 27.6–57.5) of iodine per scan would be sufficient to achieve 50 HU for each patient in the study population. This would be on average 0.486 g (SD: ±0.0210; range: 0.44–0.53) of iodine per kilogram of TBW, which is 4019 g iodine and 13.4 litres of contrast for 102 patients (Figure 4). As an example, we would like to illustrate the added value of LBW for two patients weighing 80 kg with different %LBW values. The first patient had a %LWB of 35.5% and the expected amount of contrast to reach 50 HU was 35.9 g. The second patient had a %LWB of 78.5% and the expected amount of contrast to reach 50 HU was 41.9 g. Hence, there would be a difference of six grams of iodine for these patients who received 45 g of contrast and were enhanced by 63 HU and 57 HU, respectively.

## 4. Discussion

Our results showed that the highest influence on liver enhancement was of %LBW, followed by TBW. Although the mean enhancement was >50 HU in all weight groups, the spread within groups was substantial; over one-quarter of patients did not reach the 50 HU liver enhancement threshold. Those who were enhanced by <40 HU were nearly all heavyweight patients of 90 kg or heavier, while patients enhanced by >70 HU were mostly patients weighing less than 70 kg. This indicates that our current protocol based on three weight categories overadministers contrast in lightweight patients and underadministers contrast in heavier patients. A more personalised protocol based on artificial-intelligence-determined body composition might both reduce overall contrast usage in our population and make liver enhancement more consistent between patients, but this requires prospective confirmation.

Several previous studies have investigated TBW- and LBW-adjusted contrast dosing protocols [8,18,19,20,24]. Heiken et al. [8] suggested the use of 0.521 g of iodine per kilogram of TBW for a 50 HU liver enhancement, while Kondo et al. [19] indicated that the use of LBW rather than TBW served better to achieve a consistent enhancement with reduced patient-to-patient variability. They suggested using an amount of 0.642 g of iodine per kilogram of LBW, based on a hepatic enhancement of 50 HU and a fixed average body fat percentage. Our finding supports the findings of Kondo et al. [19] and other studies [18,20,24]. However, both the studies of Kondo et al. [19] and Matsumoto et al. [24] concluded that LBW-based protocols best perform in the normal and high weight/BMI groups. In contrast, we found that LBW played the most important role in the weight groups ≤ 70 kg and 70–90 kg, wherein the spread of gI/TBW was the highest. For the group ≥ 90 kg, there was only a minor spread, and thus LBW played a less important role in this group in our study.

The differences between our results and the above-mentioned studies could be explained by the fact that the population in the study by Kondo et al. [19] was only partially comparable to our population. Our population represented a broader weight spectrum, with a range of 54–126 kg and with a median just above 80 kg, whereas in the study by Kondo et al. [19] the study population had a TBW range of 30–80 kg, with a mean just above 50 kg. Our study might, thus, have implications for a population with a wider range in weight. Reassuringly, we found the same results in the overlapping parts of the studies by Kondo et al. [19] and Matsumoto et al. [20]; LBW might be a better variable to determine the amount of iodine contrast used for light and average weight patients.

Contrast administration based on LBW might be economically effective. There was a difference of 3.2 g between the mean iodine dose per scan in the formula based on LBW and the mean iodine dose used in our current protocol. Based on the 600,000 yearly abdominal CT scans performed in the Netherlands, the new personalised method could save 1.8 tonnes of iodine a year [25], which is approximately €580,000 of yearly savings. Moreover, despite the conclusion that LBW performs better in personalising contrast application and the fact that the implementation of this finding might be beneficial if replicated prospectively, we conclude that the influence of LBW is minor. Similar to Kondo et al. [19], our equation is based on both TBW and LBW (the opposite of body fat percentage), and when dissecting the formula based on LBW (gI = ΔHU/(10.3/LBW + 82.3/TBW)) we find that TBW is still the most important factor and that LBW only has less influence.

In the study by Kondo et al. [19], an average body fat percentage of 23% was used for every patient to perform analysis, whilst some patients in their population had body fat percentages of up to 50%. We used per-patient calculated body composition for analysis. For the calculation of %LBW, we were able to use an artificial intelligence algorithm that automatically calculates body composition based on CT slices [22]. The tool proved useful for determining the body composition values for the large quantity of patients in our study, especially because this process was fully automated. 

In the literature, several methods have been used to estimate the LBW (e.g., methods proposed by James [26], Boer [27], and Janmahasatian [28]), yet no consensus has been reached on a golden standard. Therefore, our artificial intelligence tool [22] may have wide implications in measuring LBW rather than in estimating LBW. In a clinical scenario, the tool can be used in protocols containing unenhanced or arterial phase scans. If the protocol does not contain unenhanced or arterial phase scans, the body composition can be determined in several ways: from earlier recorded scans or by performing one single slice through the abdomen before scanning (as done for bolus timing acquisitions). Furthermore, bolus tracking slices may be (re)used in the future when the algorithm is tested on such arterial slices. However, the latter still has to be evaluated in future research. Moreover, while this study addresses abdominal scans, the AI algorithm can segment neck, chest, pelvis, or lower extremity scans as well when acquired, to calculate the body composition without the use of the abdomen. Once validated, the benefit could extend to those regions as well.

The limitations of this study are that this is a retrospective study design using a limited number of patients. As we needed to calculate enhancement, regular abdominal CT could not be included. Secondly, there were two outliers with enhancement levels <40 HU. The low enhancement could be due to small contrast extravasation, although this was not recorded. Another explanation could be a poor cardiac output, which results in poor enhancement and image quality [17]. However, we used premonitoring for contrast timing in our scan protocol and no scans were found to be non-diagnostic based on the timing of the scan.

Future prospective studies could investigate the impact of personalised dosing on liver enhancement and diagnostic properties, which should also take tube voltage into account [14]. Many studies already investigated the potential of low kVp settings (e.g., 70, 80, and 100 kVp) [2,14,29,30,31,32] or virtual monochromatic imaging with low kV reconstruction [33,34,35,36,37] in combination with a reduced amount of injected iodine in a more lightweight population using CT angiography protocols, wherein only the signal during the first pass of contrast media is crucial [29]. However, this has not properly been investigated for abdominal protocols yet, which rely on longer contrast media boluses to provide homogeneous enhancement of parenchymal organs, such as the liver. With the newest CT technologies (e.g., automated kVp selection, monochromatic data reconstruction, and iterative reconstruction), it is expected that more CT scans will be performed using lower kVp settings in the future [38]. As lower kVp/kV settings result in higher attenuation values, there is an opportunity to save even more contrast media than the above-mentioned €580,000. We anticipate that personalised contrast dosing is at least partly additional to the above-mentioned technological innovations.

## 5. Conclusions

In summary, in this study, we investigated the relationship between body parameters, such as TBW, LBW, and BMI, on liver enhancement in CT. We found that contrast-enhanced CT values of 40 HU and higher were of diagnostic value when assessed visually. Our data suggest the use of an artificial intelligence body composition-based algorithm to determine LBW can reduce interpatient variability in liver enhancement whilst saving contrast media. The automated nature of the algorithm makes real-time personalisation of contrast dosing technically feasible. Further research should focus on how to integrate body-composition-based personalised contrast dosing with lower tube voltage settings or monochromatic imaging.

## Figures and Tables

**Figure 1 jpm-11-00159-f001:**
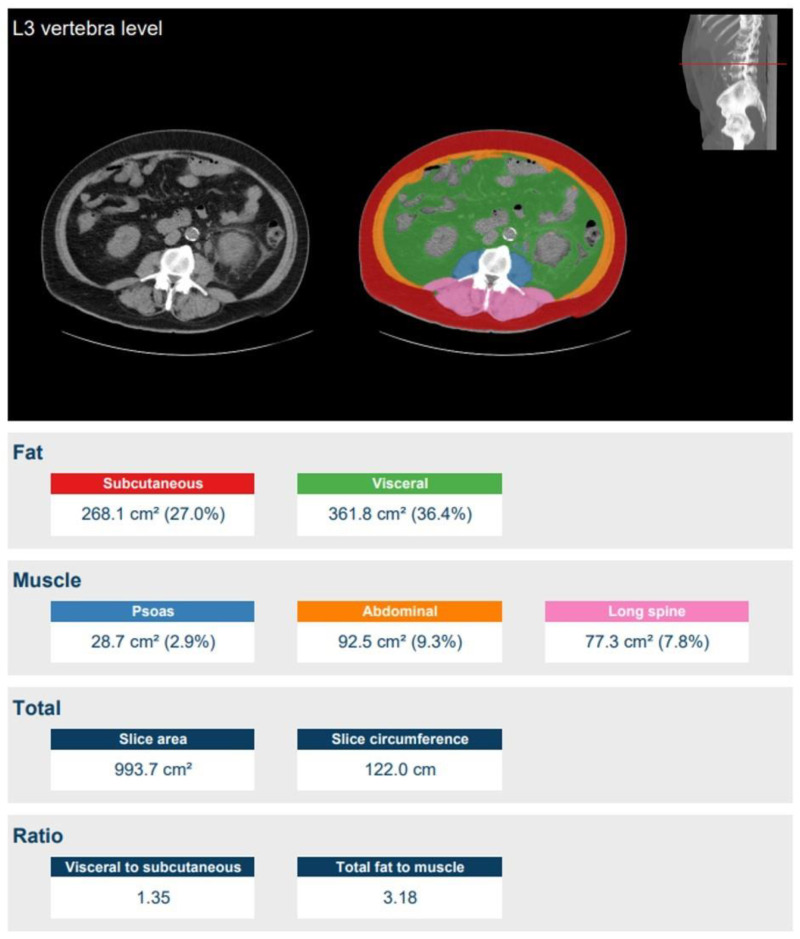
Fully automatic measurement of body composition at the lumbar 3 level [22]. Lean body weight (LBW) was defined as the difference between body weight and body fat weight, expressed in kilograms. In this example, LBW is 36.6% of the total body weight (100%—27.0% (subcutaneous fat)—36.4% (visceral fat) = 36.6% (LBW)).

**Figure 2 jpm-11-00159-f002:**
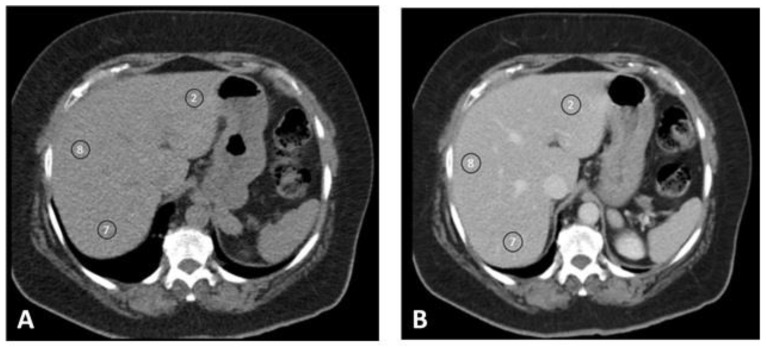
Region of interest (ROI) placement according to the Couinaud segmental classification to measure liver enhancement. ROIs were drawn in S2, S8, and S7 of the liver (when available) in unenhanced and enhanced images (portal venous phase). The degree of enhancement (ΔHU) was calculated by subtracting the unenhanced enhancement values (**A**) from enhanced enhancement values (**B**).

**Figure 3 jpm-11-00159-f003:**
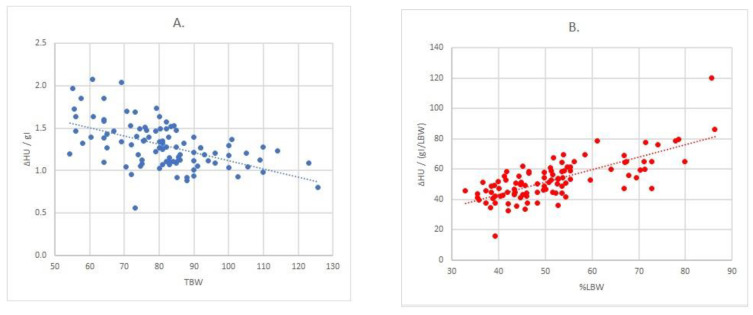
Regression analysis between enhancement and body size measures: (**A**) relationship between ΔHU/gI and TBW (*r* = 0.531; R^2^ = 0.282; *p <* 0.001); (**B**) relationship between ΔHU/(gI/LBW) and %LBW (*r* = 0.733; R^2^ = 0.538; *p <* 0.001). Note: TBW: total body weight; LBW: lean body weight.

**Figure 4 jpm-11-00159-f004:**
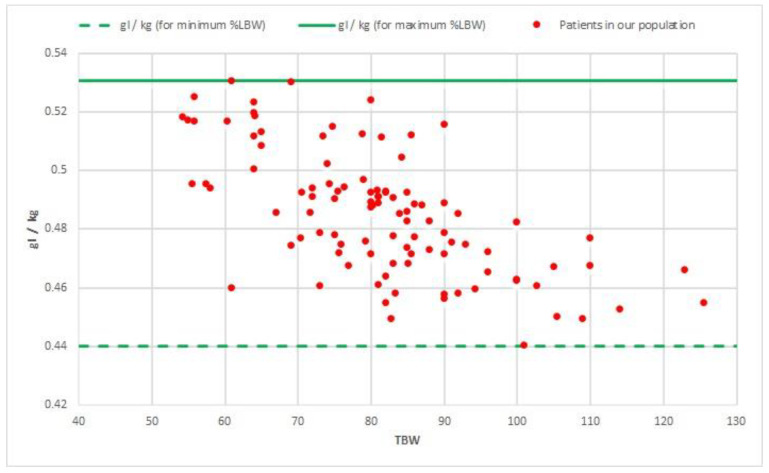
Analysis of future contrast applications: grams of iodine (gI) per kilogram (kg) TBW in the LBW formula; grams of iodine per kilogram of TBW (total body weight) in the LBW (lean body weight) formula for the population of our study. The grams of iodine per kilogram of TBW all lay between the patient with the maximum LBW (iodine (gr) maximum LBW/kg) and the patient with the lowest LBW (iodine (gr) minimum LBW/kg). Herein, the highest spreads were found in the ≤70 kg and 70–90 kg weight groups.

**Table 1 jpm-11-00159-t001:** Baseline characteristics. Normally distributed data are given as means with ±SDs and non-parametric data are given as medians with interquartile ranges (IQRs). TBW = total body weight; LBW = lean body weight in kilograms or percentage of lean body weight; BMI = body mass index.

Characteristic	Group ≤ 70 kg	Group 70–90 kg	Group ≥ 90 kg	Total
No participants	20	62	20	102
Sex male	45.0%	75.8%	80.0%	70.6%
Age (year)	70 (59–76)	69 (56–74)	64 (59–73)	68 (57–74)
TBW (kg)	62.5 (56.3–64.8)	81.0 (75.8–85.0)	101 (94.7–110)	81.0 (72.8–90.0)
LBW (kg)	40.8 (32.2–46.2)	40.8 (34.9–44.0)	41.4 (38.9–45.9)	41.1 (35.8–44.1)
%LBW	69.6 (55.3–73.8)	51.0 (43.8–53.7)	40.5 (37.5–44.5)	49.8 (42.1–55.3)
Height (cm)	168 (±13.1)	176 (±8.02)	180 (±10.9)	176 (±9.11)
BMI	21.3 (±2.01)	26.3 (±2.47)	31.5 (±4.10)	26.3 (±4.18)
Grams of iodine (mean)	38.7 (±3.88)	42.6 (±3.62)	46.3 (±3.96)	42.6 (±4.42)
Grams of iodine (median)	36.0 (36.0–43.5)	45.0 (39.0–45.0)	45.0 (45.0–45.7)	45.0 (39.0–45.0)
Grams of iodine/TBW	0.632 (±0.693)	0.530 (±0.534)	0.453 (±0.060)	0.532 (±0.081)
Grams of iodine/LBW	1.00 (±0.281)	1.07 (±0.176)	1.12 (±0.139)	1.07 (±0.196)
**Mean (± SD) or Median (IQR)**				

**Table 2 jpm-11-00159-t002:** Enhancement in liver segments for the weight groups.

Enhancement	Group ≤ 70 kg	Group 70–90 kg	Group ≥ 90 kg	Total	*p*-Value
S2 blanco	60.5 (±5.77)	56.7 (±5.02)	53.6 (±6.30)	56.8 (±5.83)	0.000
S2 PV	120.6 (±11.6)	109.8 (±11.7)	105.7 (±9.86)	111.1 (±12.3)	0.000
S2 SD	9.57 (±1.43)	11.1 (±1.89)	12.1 (±1.91)	11.0 (±1.92)	0.000
Δ S2	60.0 (±10.6)	53.1 (±10.7)	52.1 (±6.73)	54.3 (±10.3)	0.014
S8 blanco	60.7 (±5.24)	55.7 (±5.84)	51.0 (±6.61)	55.7 (±6.61)	0.000
S8 PV	120.9 (±14.2)	109.2 (±10.9)	104.4 (±9.52)	110.5 (±12.5)	0.000
S8 SD	9.19 (±0.981)	10.2 (±1.47)	11.2 (±2.18)	10.3 (±1.75)	0.000
Δ S8	60.1 (±12.6)	53.5 (±10.8)	53.4 (±8.18)	54.8 (±10.9)	0.043
S7 blanco	59.5 (±5.56)	54.5 (±5.35)	50.8 (±6.82)	54.7 (±6.32)	0.000
S7 PV	118.7 (±10.8)	107.8 (±10.1)	103.1 (±8.34)	109.0 (±11.1)	0.000
S7 SD	9.29 (±1.10)	10.6 (±1.70)	12.2 (±2.35)	10.7 (±2.09)	0.000
Δ S7	60.1 (±12.6)	53.3 (±9.25)	52.4 (±7.45)	54.3 (±9.30)	0.022
Δ S2 Δ S8 Δ S7	0.667	0.939	0.520	0.114	
Mean Δ	60.7 (±12.4)	53.3 (±9.75)	52.6 (±6.63)	54.6 (±10.2)	0.007 *
**Mean (± SD)**					

Enhancement values for the liver segments S2, S8, and S7 for the different weight groups. Values are given as means with ± SDs; *p*-values are calculated using a one-way ANOVA or repeated measures ANOVA. The blanco scans are non-enhanced scans, PV scans are scans made in the portal venous phase, the SD is given for the mean region of interest (ROI) SD, and lastly the mean enhancement is given; ΔS2 ΔS8 ΔS7 is the significance of enhancement between the three liver segments. The mean enhancement (mean ΔHU) is the average of ΔS2 ΔS8 ΔS7. Note: * Post hoc analysis showed a significant difference between ≤70 kg and 70–90 kg weight categories and between ≤70 kg and ≥90 kg weight categories.

## Data Availability

The data presented in this study are available on request from the corresponding author. The data are not publicly available due to ongoing unpublished research.

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
