# Peer review of "Towards Personalised Contrast Injection: Artificial-Intelligence-Derived Body Composition and Liver Enhancement in Computed Tomography"

_jpm, 2021, doi:10.3390/jpm11030159_

Round 1
Reviewer 1 Report
The authors conducted a retrospective study to assess the the impact of TBW, LBW, and BMI on liver enhancement in multiphase abdominal CT. The results recommended a personalized injection based on %LBW. This study put forward a relevant insight into CT inspection for liver enhancement. Nevertheless, the following points should be considered to enable the publication of manuscript.
Major point:
The author emphasized "Artificial Intelligence" (AI) in the title and the keywords. Yet, the description on the rationale and the procedure with regard to AI is surprisingly scarce. I assume AI issue should be better addressed in Introduction, Method, and Discussion if the authors would like to regard AI as an essential element in this manuscript.
Minor points:
- There should be a space between the number and the unit. Such as "300 mgI/mL", " 26 kg/m2", "90 kg", and more. There are many such items scattering in Materials and Methods, Results, and Discussion to be corrected.
- Page7, Line10: "might may" is supposed to be a typo.
Reviewer 2 Report
Dear Authors,
The review comments is to be found in the attached document.

Reviewer 3 Report
A revision of the punctuation throughout the text is needed (see some suggestions in the attached file)
Some sentences (see the attached file) need to be rewritten to clarify the meaning
Add spaces between numbers and units of measure
Abstract
Please explain all abbreviations (CT, ROI, HU, TBW, LBW, BMI...)
Introduction
See minor changes in the attached file
Add some considerations on the presence of existing studies in the literature dealing with this topic and underline novel aspects of this study.
M&M
Contrast material injection and CT protocols
The authors did not explain the rationale of the different chosen injection protocols
"In some cases, technicians adapted..." do you mean that some patients received an injection protocol different from the ones described above?
Rewrite the statistical analysis section merging the section "logistic regression" and "linear regression"
Other minor comments/revisions in the attached file

Round 2
Reviewer 1 Report
The authors have addressed all my points. I recommend the the revised version is suitable for acceptance.